# Curcumin, an Inhibitor of p300-HAT Activity, Suppresses the Development of Hypertension-Induced Left Ventricular Hypertrophy with Preserved Ejection Fraction in Dahl Rats

**DOI:** 10.3390/nu13082608

**Published:** 2021-07-29

**Authors:** Yoichi Sunagawa, Masafumi Funamoto, Kana Shimizu, Satoshi Shimizu, Nurmila Sari, Yasufumi Katanasaka, Yusuke Miyazaki, Hideaki Kakeya, Koji Hasegawa, Tatsuya Morimoto

**Affiliations:** 1Division of Molecular Medicine, School of Pharmaceutical Sciences, University of Shizuoka, Shizuoka 422-8526, Japan; y.sunagawa@u-shizuoka-ken.ac.jp (Y.S.); funamoto@u-shizuoka-ken.ac.jp (M.F.); s18804@u-shizuoka-ken.ac.jp (K.S.); s18410@u-shizuoka-ken.ac.jp (S.S.); nurmilasari@gmail.com (N.S.); katana@u-shizuoka-ken.ac.jp (Y.K.); y.miyazaki@u-shizuoka-ken.ac.jp (Y.M.); koj@kuhp.kyoto-u.ac.jp (K.H.); 2Division of Translational Research, Clinical Research Institute, Kyoto Medical Center, National Hospital Organization, Kyoto 612-8555, Japan; 3Shizuoka General Hospital, Shizuoka 420-8527, Japan; 4Department of System Chemotherapy and Molecular Sciences, Division of Bioinformatics and Chemical Genomics, Graduate School of Pharmaceutical Sciences, Kyoto University, Kyoto 606-8501, Japan; scseigyo-hisyo@pharm.kyoto-u.ac.jp

**Keywords:** p300, curcumin, hypertension, hypertrophy, LVH

## Abstract

We found that curcumin, a p300 histone acetyltransferase (HAT) inhibitor, prevents cardiac hypertrophy and systolic dysfunction at the stage of chronic heart failure in Dahl salt-sensitive rats (DS). It is unclear whether curcumin suppresses the development of hypertension-induced left ventricular hypertrophy (LVH) with a preserved ejection fraction. Therefore, in this study, we randomized DS (*n* = 16) and Dahl salt-resistant (DR) rats (*n* = 10) at 6 weeks of age to either curcumin or vehicle groups. These rats were fed a high-salt diet and orally administrated with 50 mg/kg/d curcumin or its vehicle for 6 weeks. Both curcumin and vehicle treatment groups exhibited similar degrees of high-salt diet-induced hypertension in DS rats. Curcumin significantly decreased hypertension-induced increase in posterior wall thickness and LV mass index, without affecting the systolic function. It also significantly reduced hypertension-induced increases in myocardial cell diameter, perivascular fibrosis and transcriptions of the hypertrophy-response gene. Moreover, it significantly attenuated the acetylation levels of GATA4 in the hearts of DS rats. A p300 HAT inhibitor, curcumin, suppresses the development of hypertension-induced LVH, without affecting blood pressure and systolic function. Therefore, curcumin may be used for the prevention of development of LVH in patients with hypertension.

## 1. Introduction

Heart failure (HF) is the final stage of cardiovascular diseases and the mortality and morbidity of severe HF are still high [1,2]. Hypertension is one of major risk factors for congestive HF and causes left ventricular hypertrophy (LVH) [3]. Chronic pressure and volume overload initially induce LVH with preserved cardiac function. LVH is a compensatory reaction, which eventually lead to maladaptive LVH and pathological remodeling with cardiac dysfunction [3]. This process causes the development of HF. Hypertensive patients with LVH present a high risk of developing other cardiovascular events, including HF [4]. Although effective blood pressure control with the help of agents, such as angiotensin-converting enzyme inhibitors, angiotensin II receptor blockers and β-blockers, is the primary treatment in patients with hypertension, the overall mortality from this condition is still considerably high [5,6]. Consequently, it is important to identify more effective agents than existing medicines to develop novel pharmacological therapies for HF.

Chronic stresses, such as hypertension and myocardial infarction (MI), up-regulated cellular signaling pathways, increasing the expression levels of p300, a transcriptional co-activator and enhancing the activity of its histone acetyltransferase (HAT) enzyme in cardiomyocytes [7,8]. The activation of p300 can induce hypertrophy-responsive transcriptional factors, such as GATA4, myocyte enhancer factor 2 and serum responsive factor, change the expression levels of fetal genes, such as the atrial natriuretic factor (ANF), brain natriuretic peptide (BNP), endothelin-1 and β-myosin heavy chain (β-MHC) and ultimately lead to cardiomyocyte hypertrophy [9,10,11]. The transgenic mice with cardiac overexpression of p300 promotes the acetylation of GATA4 and augments the left ventricle (LV) remodeling after MI compared to the wild-type mice, while the hearts of the HAT-deficient mutant p300 mice did not exhibit these changes [12]. The cooperation of p300/GATA4 and acetylation of GATA4 are critical events in cardiomyocyte hypertrophy and the development of heart failure [9,13,14]. Based on these results, p300-HAT activity may be used as a novel pharmacological target for HF [15].

Curcumin (diferuloyl-methane) is a natural yellow compound isolated from the rhizome of *Curcuma longa*. Many reports demonstrate that curcumin shows different kinds of beneficial effects, including antioxidant [16], anti-inflammatory [17] and antitumor effects [18,19], while also inhibiting the p300/CBP-specific HAT activity [20]. Even though curcumin has low bioavailability, highly bioavailable forms of curcumin is expected to exhibit potential health benefits and expected to contribute to the health management [21,22]. Our previous study has shown that curcumin prevents systolic dysfunction and the development of myocardial cell hypertrophy via the inhibition of p300-HAT activity in two HF models in rats, with MI and hypertension [23]. Although curcumin is expected to be a novel therapeutic agent for HF in the clinical setting, [24,25,26] the effect of curcumin on hypertension-induced LVH with preserved ejection fraction remains unclear.

In this study, we investigated whether curcumin exhibited any beneficial effects on hypertension-induced LVH in the Dahl salt-sensitive (DS) and salt-resistant (DR) rats.

## 2. Materials and Methods

### 2.1. Dahl Salt-Sensitive Rats

All animal experiments were approved and carried out according to the Guide for the Care and Use of Laboratory Animals by the Institute of Laboratory Animal Care and Use Committee, Graduate School of Medicine, Kyoto University (KUHP13192) and the University of Shizuoka (US206455). Six-week-old male DS and DR rats were bought from Japan SLC Inc (Shizuoka, Japan). These rats were fed a low-salt diet with 0.3% sodium chloride (NaCl) for the first 6 weeks. Then, they were fed a high-salt diet with 8% NaCl until the age of 12 weeks.

### 2.2. Treatment

Curcumin powder and gum arabic were purchased from FUJIFILM Wako Pure Chemicals Corporation (Osaka, Japan). The DR and DS rats (age, 6-week-old) were randomly assigned to two groups (DR curcumin, *n* = 5; DS curcumin, *n* = 8; DR vehicle, *n* = 5; DS vehicle, *n* = 8). Rats were administered daily with curcumin (50 mg/kg/d, diluted in 1% arabic gum) and vehicle (1% arabic gum) via an oral gastric gavage. This daily oral administration of curcumin or vehicle was continued for 6 weeks, until the rats reached 12 weeks of age. Systolic and diastolic blood pressure (SBP and DBP) were evaluated by the tail-cuff method using BP-98A system (Softron Inc., Tokyo, Japan).

### 2.3. Echocardiography

Cardiac function was measured by non-invasive echocardiography according to a previously described method [27]. In brief, echocardiography was performed when the rats were under light anesthesia. Then, the LV end-diastolic dimension (LVEDD), end-systolic dimension (LVESD), interventricular septum thickness (IVST) and posterior wall thickness (PWT) were measured using M-mode tracing from the short-axis view of the LV at the level of papillary muscle. Fractional shortening (FS) was calculated as follows: %FS = [(LVEDD − LVESD)/LVEDD] × 100. LV mass index was calculated as follows: LV mass index = [1.04 × (LVEDD + IVST + PWT)^3^ − (LVEDD)^3^]. The specific gravity of the myocardium was determined to be 1.04. All measurements were performed according to the guidelines of the American Society for Echocardiology and averaged over three consecutive cardiac cycles.

### 2.4. Histological Analysis

The hearts were excised, cut, fixed in 3.7% paraformaldehyde and embedded in paraffin. The mid-level of LV sections (4 μm) were staining with hematoxylin and eosin (H&E) and Masson’s trichrome stains following standard protocols. Measurements of cross-sectional myocardial cell diameter and perivascular fibrosis were previously described [28]. The scales of coronary artery larger 50 μm in each rat were measured.

### 2.5. Quantitative Reverse Transcription-Polymerase Chain Reaction (qRT-PCR)

Total RNA from LV in each rat was isolated by TRIzol reagent (Invitrogen, Waltham, MA, USA). Complemental DNA was synthesized using 1 μg of RNA by a standard protocol (Applied Biosystems, Waltham, MA, USA). Primer sequences of ANF, β-MHC and glyceraldehyde 3-phosphate dehydrogenase (GAPDH) have been described previously [29]. The mRNA expression levels were quantified using SYBR Green (Applied Biosystems, Waltham, MA, USA). The values were normalized to GAPDH as a reference gene and expressed relative to the control.

### 2.6. Immunoprecipitation and Immuno-Blotting

Nuclear extracts (NE) prepared from the heart tissue in each rat were previously described [30]. In brief, Each NE were subjected to immunoprecipitation with goat polyclonal anti-GATA4 antibody (sc-1237, Santa Cruz Biotechnology, Dallas, TX, USA). NE and immunoprecipitated samples were subjected to immune-blotting using mouse monoclonal anti-GATA4 (sc-25130), rabbit polyclonal anti-p300 (sc-584) (Santa Cruz Biotechnology, Dallas, TX, USA), rabbit polyclonal anti-acetyl-lysine (#9441, Cell Signaling, Danvers, MA, USA) and mouse monoclonal anti-β-actin (A2228, Sigma-Aldrich, St. Louis, MO, USA) antibodies. The signal density was detected using Amersham Imager 680 (Cytiva, Marlborough, MA, USA), quantified by Image J v1.47 and corrected with β-actin.

### 2.7. Statistics

Power analysis was performed using G*Power 3.197 software [31]. Based on our previous study [23], we set an effect size of 0.4, a power of 0.8, statistical significance of 0.05 and expected attrition of 10% and determined that a sample size was ten rats per DS group. The data are expressed as the mean ± standard error (SE). Data analysis were performed using the StatView 5.0 software by two-way analysis of variance, following by Tukey’s multiple-comparison test. A value of *p* < 0.05 is statistical significance.

## 3. Results

### 3.1. The Effect of Curcumin on Hemodynamics

To investigate the effect of curcumin therapy on hypertension, we utilized DS and DR rats. Both rats (6 weeks old) fed an 8% NaCl diet for 6 weeks, developing hypertension and concentric LVH [32]. Before treatment, body weight (BW), SBP, DBP, heart rate (HR) and echocardiographic parameters were not different between both groups (Table 1). SBP and DBP were weekly measured (Figure 1). At 12 weeks of age, SBP and DBP were significantly high in DS rats compared to in DR rats (Table 2). These values were not different between both groups.

### 3.2. The Effect of Curcumin on Echocardiographic Parameters and Cardiac Function

To determine the effects of curcumin on concentric LVH and cardiac function, we performed physiological analysis in both rats at 12 weeks of age. The representative images of DS vehicle and DS curcumin groups are shown in Figure 2A. Hypertension significantly reduced LVEDD in the DS vehicle group (6.96 ± 0.18 mm, *p* < 0.001) compared to the DR vehicle group (8.93 ± 0.17 mm) (Figure 2B). Curcumin treatment did not affect LVEDD in DR (8.75 ± 0.49 mm) and DS rats (7.75 ± 0.29 mm). FS as an indicator of systolic function showed no difference between each group (DR vehicle: 49.7 ± 2.8%, DR curcumin: 52.0 ± 3.5%, DS vehicle: 58.8 ± 1.7%, DS curcumin: 57.6 ± 1.6%) (Figure 2C).

LVH is defined by the measurement of the PWT and the calculated LV mass index [29,30]. PWT was significantly thicker in DS vehicle group (2.03 ± 0.04 mm, *p* < 0.01) than that in DR vehicle group (1.02 ± 0.08 mm) (Figure 2D). DS rats significantly increased LV mass index (1.07 ± 0.05 g, *p* < 0.05) compared to DR rats (0.86 ± 0.04 g). Curcumin treatment significantly reduced the LV mass index in DS rats (0.90 ± 0.06 g, *p* < 0.05).

While the high-salt diet reduced body weight (BW) in DS rats, these values were also similar in DS vehicle and DS curcumin groups. Hypertension induced an increase in heart weight (HW), LV weight (LVW), the HW/BW ratio and the LVW/BW ratio. Curcumin treatment significantly reduced these parameters without affecting blood pressure in DS rats. These results indicated that curcumin treatment significantly prevented hypertension-induced LVH without affecting blood pressure and cardiac function in DS rats.

### 3.3. The Effects of Curcumin on Myocardial Cell Hypertrophy and Fibrosis

To investigate whether curcumin affected hypertension-induced myocardial cell hypertrophy and fibrosis, we performed histological analysis in both rats. The representative images of H&E stained of transversely sectioned LV and myocardial cells in each group are shown in Figure 3A,B. The data of myocardial cell diameter in each group are shown in Figure 3C. Microscopic analysis revealed that hypertension induced an increase in myocardial cell diameter in DS vehicle group compared to that in DR vehicle group (lanes 1 and 3, *p* < 0.001). Curcumin treatment significantly reduced this increase compared to vehicle treatment in DS rats (lanes 3 and 4, *p* < 0.01). The representative images of perivascular fibrosis in each group are shown in Figure 3D. The ratio of perivascular fibrosis area around the coronary artery was significantly increased in DS vehicle group compared to in DR vehicle group (Figure 3E, lanes 1 and 3, *p* < 0.001). Hypertension-induced perivascular fibrosis was significantly reduced by curcumin treatment (lanes 3 and 4, *p* < 0.01). The areas of interstitial fibrosis in each group were comparable (Appendix A). These indicated that curcumin treatment significantly reduced hypertension-induced increases in myocardial cell diameter and perivascular fibrosis in DS rats.

### 3.4. The Effects of Curcumin on the Transcriptions of Hypertrophy-Response Genes

The development of LVH correlated with the upregulations of hypertrophy-response gene transcriptions, including ANF and β-MHC. We investigated ANF and β-MHC mRNA levels of LV tissues in both rats. Hypertension significantly enhanced ANF and β-MHC gene transcriptions (Figure 4A,B, lanes 1 and 3, *p* < 0.05). Curcumin treatment significantly suppressed these activations (lanes 3 and 4, *p* < 0.05).

### 3.5. The Effects of Curcumin on the Acetylated Levels of GATA4

A transcriptional coactivator, p300, mediated GATA4 acetylation is one of the critical events for the upregulation of GATA4-dependent hypertrophy-response gene transcriptions and myocardial cell hypertrophy [12,13]. We determined whether LVH with hypertension was associated with the acetylation of GATA4 in vivo. Western blotting analysis demonstrated that the protein level of p300 was higher in DS rats than in DR rats (Figure 5A,B). Curcumin treatment did not alter p300 expression. The protein level of GATA4 did not differ between both groups (Figure 5A,C). To detect the acetylated levels of GATA4, same NE was subjected to immunoprecipitation with anti-GATA4 antibody were performed. We confirmed the acetylated form of GATA4 was increased in DS rats at the 12 weeks-old (Figure 5D,E, lanes 1 and 3, *p* < 0.05). Notably, hemodynamic overload-induced acetylation of GATA4 was reduced by curcumin treatment in DS rats (lanes 3 and 4, *p* < 0.05). Based on these results, curcumin suppressed hypertension-induced LVH and hypertrophy-response gene transcription through the inhibition of GATA-acetylation.

## 4. Discussions

We examined the effects of a p300-HAT inhibitor, curcumin, on LVH and cardiac function at the onset of hypertension, using DS and DR rats. Curcumin did not influence hypertension and cardiac function but significantly suppressed the hypertension-induced LVH. In addition, curcumin significantly suppressed the expression levels of the hypertension-induced cardiac hypertrophic marker genes as well as the acetylation of GATA4. From the above results, it was shown that curcumin may be useful for the prevention of hypertension-induced LVH with conserving ejection fraction.

Curcumin, a natural compound, exhibits therapeutic and beneficial potency on cardiovascular diseases, including hypertension. Yao et al. showed that curcumin treatment (300 mg/kg/d) reduced angiotensin II-induced hypertension in mice [32]. N (ω)-nitro-L-arginine methyl ester (L-NAME)-induced hypertension was decreased by curcumin (50 and 100 mg/kg/d) [33]. However, Muta et al. showed that curcumin ameliorates nephrosclerosis without affecting hypertension in Dahl rats [34]. Our previous and recent studies also confirmed that curcumin had no significant effect on blood pressure in Dahl rats. Because curcumin has potent antioxidant properties that effectively scavenge reactive oxygen species (ROS), curcumin may exhibit a hypotensive action on ROS-related increase in blood pressure [32,33]. In contrast, the salt-sensitive hypertension in Dahl rats is based on the complex mechanisms [35,36]. An antioxidant, resveratrol, exhibit the therapeutic effect for cardiac dysfunction without affecting blood pressure and cardiac hypertrophy in Dahl rats [37]. Tempol was no effect on hypertension in young Dahl rats [38]. It seems that only antioxidant activity cannot correct salt-sensitive hypertension in Dahl rats. However, curcumin has a protective function against LVH, cardiac dysfunction and renal dysfunction independent of hypertension, indicating that curcumin in addition to antihypertensive agents such as ARB and ACEI may exhibit synergistic benefits in hypertension-related diseases [25].

The protein level of p300 increased in LV of ischemic, dilated, or unspecified end-stage cardiomyopathy compared to non-failing heart [39]. The ubiquitin-proteasome pathway downregulates p300 protein levels via the deacetylation of lysine residues for ubiquitination by SIRT1 [40]. However, it remains unclear how the levels of p300 are upregulated in hearts with pathological hypertrophy. In our previous and recent studies, we confirmed that hemodynamic overload induced an increase in myocardial p300 levels and acetylation of GATA4 at the stage of not only CHF but also LVH in Dahl rats [23]. Our data also showed that p300 HAT inhibition by curcumin treatment did not alter the p300 protein levels, but prevented the acetylation of GATA4 in both stages. These results suggest that the increase in p300 might be independent of HAT activity and associated with other hypertrophic pathways, such as ERK1/2, MAPK38 and Akt [41,42].

Several studies have implicated that histone de-acetylases (HDAC) and HAT controls cardiac hypertrophy [43]. Class IIa HDAC5 and HDAC9 are associated with MEF2, another acetylation target of p300 and regulate its transcriptional activity in response to hypertrophic stimuli. HOP/HDAC2 cooperates with GATA4 and regulates cardiomyocyte proliferation through GATA4 de-acetylation during embryonic development [44]. SIRT7 exerts an anti-hypertrophic action through the deacetylation of GATA4 [45]. In this study, we showed that curcumin reduced the acetylated level of GATA4 in DS rats at the 12-weeks-old. However, the precise mechanism of de-acetylation GATA4 during concentric and pathological hypertrophy is still unclear. Further studies will have to delineate which HDACs regulate the deacetylation of GATA4 during the development of LVH.

Zhou et al. reported that DS rats at the LVH stage exhibited a decrease in NO bioactivity, as the levels of eNOS were decreased by the production of O_2_^−^ [46]. In addition, the production of vascular ET-1 and peroxynitrite were upregulated in DS rats [47]. Oxidative stress is one of the mediators to induce myocardial fibrosis associated with cardiac hypertrophy [48]. Curcumin is also powerful anti-oxidant and anti-inflammatory compound and acts as a specific p300 HAT inhibitor [49]. Kim et al. reported that curcumin reduced TNFα-mediated inflammatory responses in human endothelial cells [50]. Mito et al. demonstrated that curcumin protected experimental autoimmune myocarditis through inhibiting cardiac inflammation [51]. It is possible that the inhibitory effects of curcumin against oxidative stress and inflammatory process may reduce hemodynamic overload-induced perivascular fibrosis. Therefore, it is important to clarify the specific effects of curcumin on cardiac fibroblasts as well as its role on HF model animals in future studies.

In this study, we used a dose of 50 mg/kg/day curcumin in Dahl rats. This high dose may be unrealistic in human study. Recently, highly absorptive forms of curcumin have been developed [21]. We also have developed two type of higher absorptive curcumin preparations, curcuRouge^TM^ and Theracurmin^®^ [22,52]. These preparations may be expected to exhibit clinical efficiency of curcumin at a low dosage. Future clinical studies are needed to examine the efficacy of highly absorptive curcumin on the prevention of LVH in patients with hypertension.

In conclusion, we have indicated the therapeutic effect of curcumin on hypertension-induced LVH with preserved ejection function through the decrease in acetylated form of GATA4 in Dahl rats. This compound may be used to prevent the development of LVH in patients with hypertension.

## Figures and Tables

**Figure 1 nutrients-13-02608-f001:**
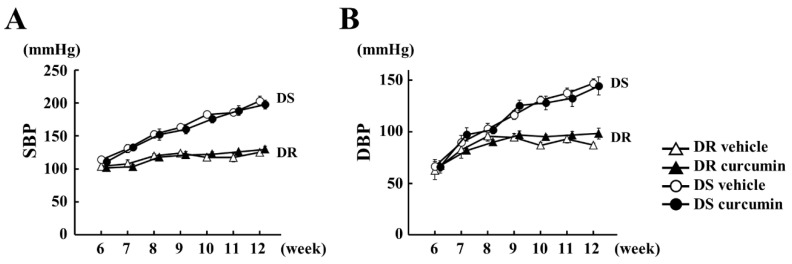
Curcumin treatment did not inhibit hypertension in Dahl salt-sensitive (DS) rats. (**A**,**B**) Blood pressure levels of DS and Dahl salt-resistant (DR) rats were examined by tail cuff methods in each group. The value expressed the mean ± standard error (SEM) from 5 DR and 8 DS rats in each group. White triangle: DR vehicle, black triangle: DR curcumin, white circle: DS vehicle, black circle: DS curcumin. SBP = systolic blood pressure. DBP = diastolic blood pressure.

**Figure 2 nutrients-13-02608-f002:**
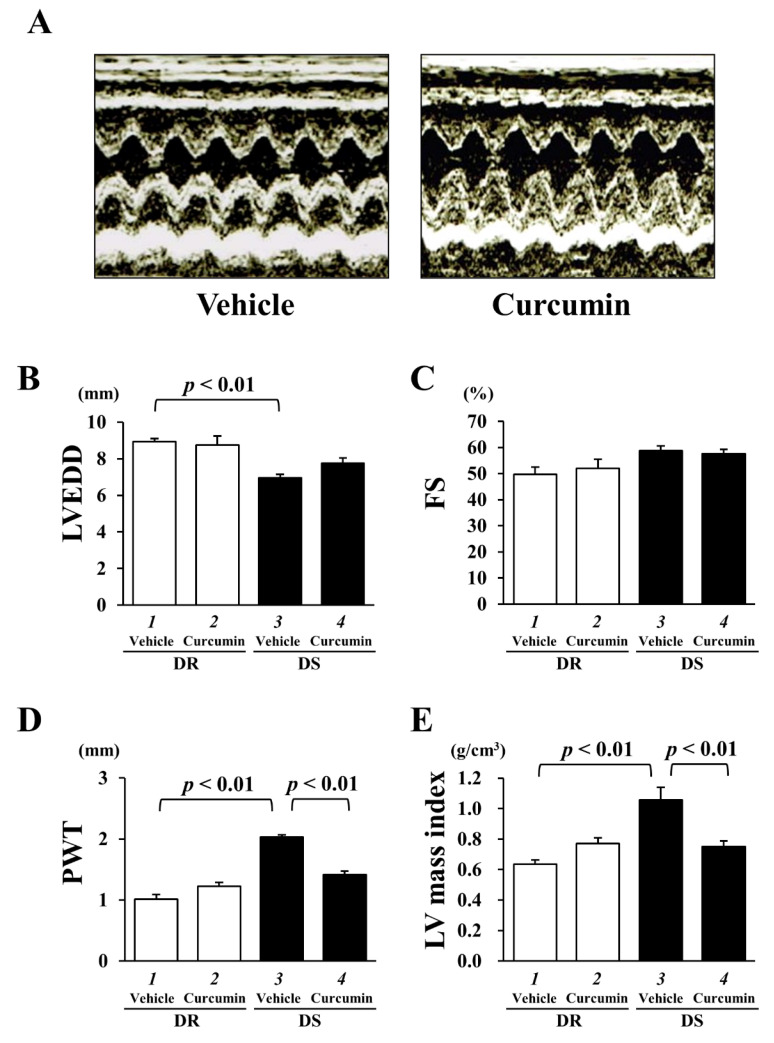
Curcumin treatment significantly reduced hypertension-induced posterior wall thickness (PWT) and left ventricle (LV) mass index without changing systolic function in DS rats. (**A**) Representative images of echocardiographic analysis in DS rats with vehicle or curcumin treatment. (**B**) LV end-diastolic dimension (LVEDD). (**C**) Fractional shortening (FS). (**D**) PWT. (**E**) LV mass index. Each value represents the mean ± SEM from 5 DR and 8 DS rats in each group.

**Figure 3 nutrients-13-02608-f003:**
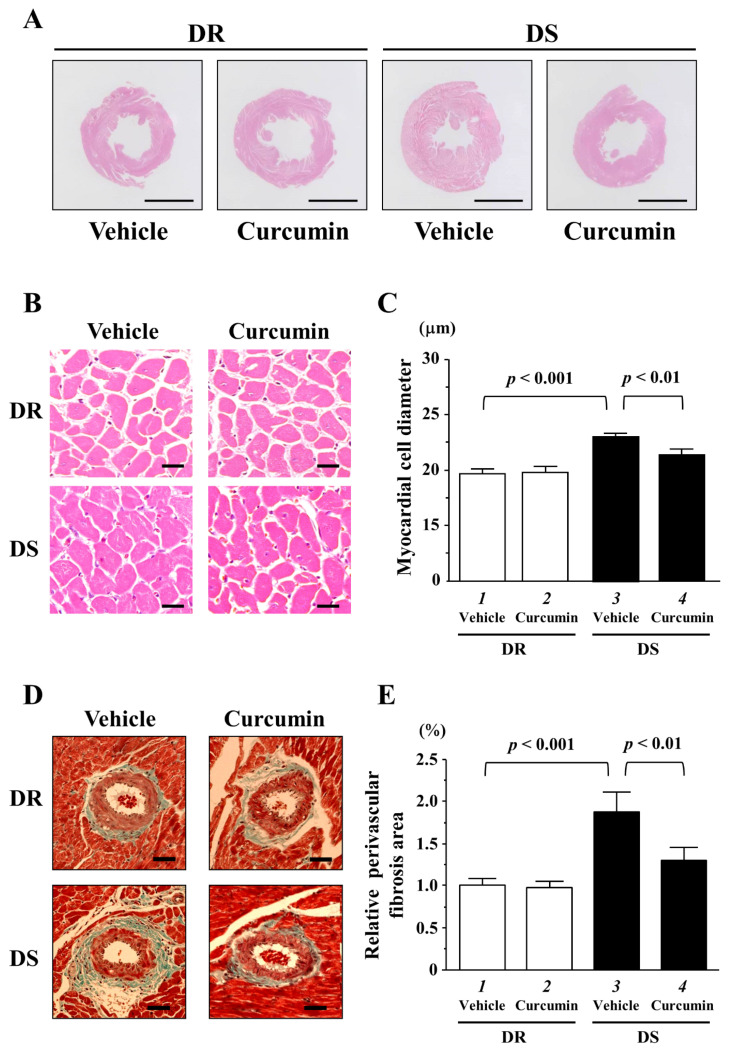
The hemodynamic overload-induced concentric left ventricular hypertrophy (LVH), myocardial cell hypertrophy and perivascular fibrosis were significantly reduced by curcumin treatment in DS rats. (**A**) The representative hematoxylin and eosin (H & E)-stained cross-sectional LV tissues in both rats with vehicle or curcumin treatment. Scale bar indicates 5 mm. (**B**) The representative images of myocardial cells with H&E staining in each rat. Scale bar indicates 20 μm. (**C**) Fifty myocardial cell diameters were measured in each rat. The data are the mean ± SEM. (**D**) The representative images of Masson trichrome-stained cross-sectional LV tissues in both rats. Scale bar indicates 50 μm. (**E**) The ratio of perivascular fibrosis area/lumen area of coronary artery was quantified. All values were normalized to DR vehicle group which was set at 1.0 as the control.

**Figure 4 nutrients-13-02608-f004:**
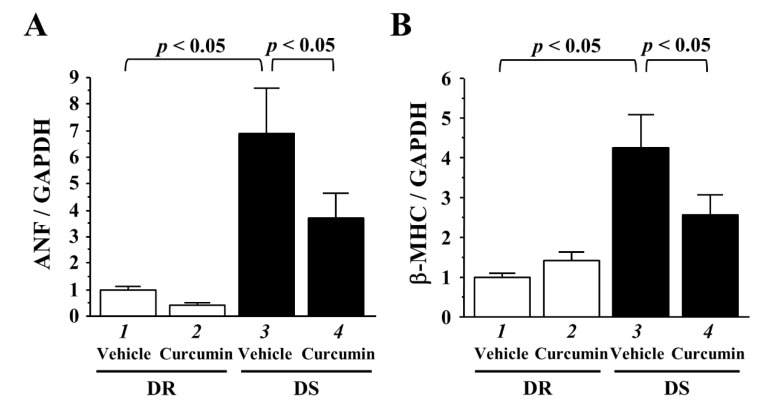
Curcumin significantly prevented the transcriptions of hemodynamic overload-induced hypertrophy-response genes in the LVs of DS rats. (**A**,**B**) The mRNA levels of atrial natriuretic factor (ANF) (**A**) and beta-myosin heavy chain (β-MHC) (**B**) of LV tissues in each rat were quantified and normalized to glyceraldehyde 3-phosphate dehydrogenase (GAPDH). All values were normalized to DR vehicle group, which was set at 1.0 as the control.

**Figure 5 nutrients-13-02608-f005:**
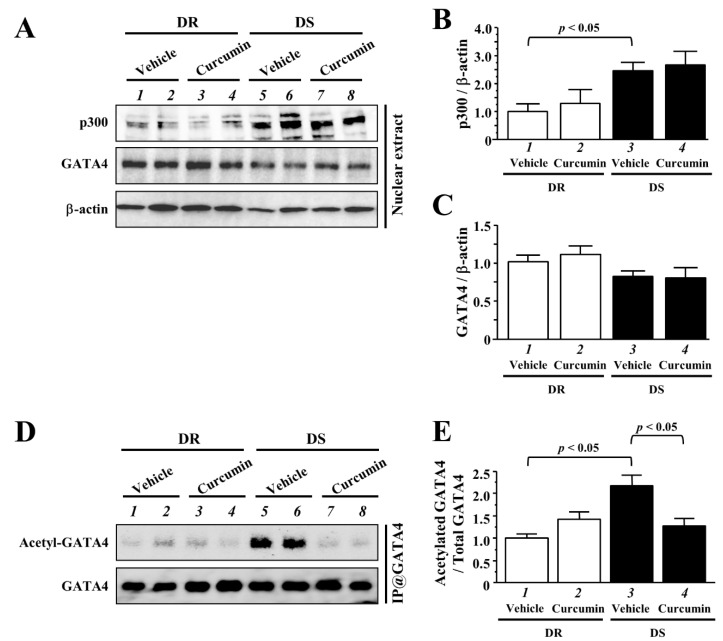
Curcumin significantly prevented hemodynamic overload-induced acetylation of GATA4 in DS rats. (**A**) Nuclear extracts (NE) of LV tissues in each rat were subjected to western blotting with indicated antibodies. (**B**,**C**) The protein levels of p300 (**B**) and GATA4 (**C**) were normalized by β-actin. (**D**) The immunoprecipitated samples with anti-GATA4 antibody were subjected to western blotting with indicated antibodies. (**E**) The acetylated level of GATA4 was normalized to total GATA4. The band intensities were quantified using Image J v1.47 software. The values are represented as the mean ± SEM. All values were normalized to DR vehicle group, which was set at 1.0 as the control.

**Table 1 nutrients-13-02608-t001:** Hemodynamic parameters in Dahl rats at 6-week-old.

	*n*	SBP mmHg	DBP mmHg	HR/Min	LVEDD mm	PWT mm	FS %
DR vehicle	5	104 ± 1	62 ± 9	439 ± 8	7.46 ± 0.29	0.80 ± 0.06	57.6 ± 4.1
DR curcumin	5	102 ± 3	67 ± 5	426 ± 20	7.53 ± 0.20	0.88 ± 0.09	57.7 ± 2.3
DS vehicle	8	111 ± 3	67 ± 6	416 ± 22	6.86 ± 0.25	0.95 ± 0.04	61.1 ± 1.8
DS curcumin	8	111 ± 4	67 ± 6	391 ± 11	6.92 ± 0.15	1.00 ± 0.06	60.9 ± 1.7

Values are mean ± S.E. in each group. SBP = systolic blood pressure, DBP = diastolic blood pressure, HR = heart rate, LVEDD = Left ventricular end-diastolic dimension, PWT = posterior wall thickness, FS = fraction shortening.

**Table 2 nutrients-13-02608-t002:** BW, HW, LVW, and blood pressure parameters in Dahl rats at 12-weeks-old.

	BW g	HW g	LVW mg	HW/BW mg/g	LVW/BW mg/g	SBP mmHg	DBP mmHg	HR /min
DR vehicle	376 ± 10	1.09 ± 0.03	809 ± 29	2.91 ± 0.08	2.16 ± 0.10	125 ± 3	89 ± 3	390 ± 16
DR curcumin	378 ± 8	1.11 ± 0.03	817 ± 17	2.93 ± 0.04	2.16 ± 0.06	129 ± 5	96 ± 6	378 ± 13
DS vehicle	319 ± 13 *	1.24 ± 0.03 *	965 ± 39 *	3.91 ± 0.10 *	3.05 ± 0.11 *	203 ± 7 *	147 ± 4 *	389 ± 10
DS curcumin	311 ± 6 *	1.12 ± 0.03 ^#^	873 ± 28 ^#^	3.62 ± 0.10 ^#^	2.68 ± 0.11 ^#^	198 ± 6 *	139 ± 9 *	398 ± 9

Values are mean ± S.E. in each group. BW = Body weight, HW = Heart weight, LVW = Left ventricle weight. * *p* < 0.05 vs DR vehicle, # *p* < 0.05 vs DS vehicle.

## Data Availability

The data in this study are available on request from the corresponding authors.

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
