# Peer review of "Curcumin, an Inhibitor of p300-HAT Activity, Suppresses the Development of Hypertension-Induced Left Ventricular Hypertrophy with Preserved Ejection Fraction in Dahl Rats"

_nutrients, 2021, doi:10.3390/nu13082608_

Round 1

Reviewer 1 Report

Great work. This reviewer felt that the authors should pay attention to the following. This manuscript has many 'overstatements' regarding curcumin effects. This reviewer would like to suggest the authors to be a bit more careful in making many of the statements. For example, prevention vis reduction vs alleviation are 3 different things. The other issue is "how long and what dose is good" for humans is controversial. Normally, people use turmeric in many forms which contains substantial amounts of curcumin but no one knows disease-specific doses!

1. Abstract: Last 2 lines: [Therefore, curcumin may be used for the treatment of hypertensive patients with preserved ejection fractions.]{This can not be suggested based on a rat study- consider a modified statement!}

2. Page 2: Lines 76-86 [Important references missing! Consider including as appropriate.

https://www.mdpi.com/1420-3049/25/6/1397

https://pubmed.ncbi.nlm.nih.gov/22272768/

3. Explain specific situations when LVH develop. Also include in intro which other antioxidants were used to alleviate LVH!

4. Surprise! surprise!!?? [Page-3: Lines 107- 111 reads "This daily oral administration of curcumin or vehicle was continued for 6 weeks, until the mice reached 12 weeks of age. Systolic blood pressure (SBP) and diastolic blood pressure (DBP) were measured weekly by the tail-cuff method using an indirect blood pressure system (BP-98A; Softron Inc., Japan)."] {The title of the MS reads Rats??}

5. Page 12: Lines 370-371: [This compound may be used to prevent the development of cardiovascular diseases in patients with hypertension.] [Curcumin's ability to reduce CV diseases is not new in the literature. The real question is - does it delay or prevent? One has to be realistic in using terms.}

Author Response

To Review1

Comment:

Great work. This reviewer felt that the authors should pay attention to the following. This manuscript has many 'overstatements' regarding curcumin effects. This reviewer would like to suggest the authors to be a bit more careful in making many of the statements. For example, prevention vis reduction vs alleviation are 3 different things. The other issue is "how long and what dose is good" for humans is controversial. Normally, people use turmeric in many forms which contains substantial amounts of curcumin but no one knows disease-specific doses!

Response:

Thank you very much for your valuable comments on our manuscript. We have revised our manuscript according to your suggestion:

Comment:

  1. Abstract: Last 2 lines: [Therefore, curcumin may be used for the treatment of hypertensive patients with preserved ejection fractions.]{This can not be suggested based on a rat study- consider a modified statement!}

Response:

                 Thank you for your suggestion. We corrected this abstract in the revised manuscript as follows:

Line 23-40

Abstract: We found that curcumin, a p300 histone acetyltransferase (HAT) inhibitor, prevents cardiac hypertrophy and systolic dysfunction at the stage of chronic heart failure in Dahl salt-sensitive rats (DS). It is unclear whether curcumin suppresses the development of hypertension-induced left ventricular hypertrophy (LVH) with a preserved ejection fraction. Therefore, in this study, we randomized DS (n = 16) and Dahl salt-resistant (DR) rats (n = 10) at 6 weeks of age to either curcumin or vehicle groups. These rats were fed a high-salt diet and orally administrated with 50 mg/kg/d curcumin or its vehicle for 6 weeks. Both curcumin and vehicle treatment groups exhibited similar degrees of high-salt diet-induced hypertension in DS rats. Curcumin significantly decreased hypertension-induced increase in posterior wall thickness and LV mass index, without affecting the systolic function. It also significantly inhibited hypertension-induced increases in myocardial cell diameter, perivascular fibrosis, and transcriptions of the hypertrophy-response gene. Moreover, it significantly attenuated the acetylation levels of GATA4 in the hearts of DS rats. A p300 HAT inhibitor, curcumin, suppresses the development of hypertension-induced LVH, without affecting blood pressure and systolic function. Therefore, curcumin may be used for the prevention of development of LVH in patients with hypertension.

Comment:

  1. Page 2: Lines 76-86 [Important references missing! Consider including as appropriate.

https://www.mdpi.com/1420-3049/25/6/1397

https://pubmed.ncbi.nlm.nih.gov/22272768/

Response:

              Thank you for your comment. We added and referred above papers in the revised manuscript.

Line 76-89

“Curcumin (diferuloyl-methane) is a natural yellow compound isolated from the rhizome of Curcuma longa. Many reports demonstrate that curcumin shows different kinds of beneficial effects, including antioxidant [16], anti-inflammatory [17], and anti-tumor effects [18,19], while also inhibiting the p300/CBP-specific HAT activity [20]. Even though curcumin is a low bioavailability, highly bioavailable forms of curcumin is expected to exhibit its attractive potency and to contribute to the health management [21,22]. In our previous study, curcumin prevents systolic dysfunction and the development of myocardial cell hypertrophy via the inhibition of p300-HAT activity in two heart failure models in rats, with MI and hypertension [23]. Although curcumin is expected to be a novel therapeutic agent for heart failure in the clinical setting, [24,25,26] the effect of curcumin on hypertension-induced LVH with preserved ejection fraction remains unclear.”

  1. Bulku, E.; Stohs, S.J.; Cicero, L.; Brooks, T.; Halley, H.; Ray, S.D. Curcumin exposure modulates multiple pro-apoptotic and anti-apoptotic signaling pathways to antagonize acetaminophen-induced toxicity. Curr. Neurovasc. Res. 2012, 9 (1), 58-71.
  2. Sunagawa, Y.; Miyazaki, Y.; Funamoto, M.; Shimizu, K.; Shimizu, S.; Nurmila, S.; Katanasaka, Y.; Ito, M.; Ogawa, T.; Ozawa-Umeta, H.; Hasegawa, K., Morimoto, T. A novel amorphous preparation improved curcumin bioavailability in healthy volunteers: A single-dose, double-blind, two-way crossover study. J. Funct. Foods. 2021, 81, 104443

Comment:

  1. Explain specific situations when LVH develop. Also include in intro which other were used to alleviate LVH!

Response:

                 Thank you for your suggestion. We modified the introduction to explain the development of LVH in the revised manuscript. An antioxidant compound, resveratrol, exhibit therapeutic effect for cardiac dysfunction without affecting blood pressure and cardiac hypertrophy in Dahl rats [37]. Although oxidative stress related to the development of hypertension, we mentioned that the antioxidant therapy was no effect on hypertension in Dahl rats [38]. Moreover, we focused on the p300-HAT inhibitory action of curcumin but not antioxidant activity of that in this study. And then, we found that curcumin treatment prevented hypertension-induced LVH independent hypertension in Dahl rats. Thus, we believed that p300-HAT inhibition of curcumin but not its antioxidant activity possesses the beneficial effects on LVH with preserved ejection fraction. We add the effect of other antioxidants on hypertension in Dahl rats in the discussion section of our revised manuscript.

Line 43-58

Heart failure (HF) is the final stage of cardiovascular diseases and the mortality and morbidity of severe HF are still high [1,2]. Hypertension is one of major risk factors for congestive heart failure and causes left ventricular hypertrophy (LVH) [3]. Chronic pressure and volume overload initially induce LVH with preserved cardiac function. LVH is a compensatory reaction, which eventually lead to maladaptive LVH and pathological remodeling with cardiac dysfunction [3]. This process causes the development of HF. Hypertensive patients with LVH present a high risk of developing other cardiovascular events, including HF [4]. Although effective blood pressure control with the help of agents, such as angiotensin-converting enzyme inhibitors, angiotensin II receptor blockers, and b-blockers, is the primary treatment in patients with hypertension, the overall mortality from this condition is still considerably high [5,6]. Consequently, it is important to identify more effective agents than existing medicines to develop novel pharmacological therapies for HF.”

Line 303-324

“A natural compound, curcumin, exhibits therapeutic and beneficial potency on cardiovascular diseases, including hypertension. Yao et al. showed that curcumin treatment (300 mg/kg/d) reduced angiotensin II-induced hypertension in mice [32]. N (ω)-nitro-L-arginine methyl ester (L-NAME)-induced hypertension was decreased by curcumin (50 and 100 mg/kg/d) [33]. However, Muta et al. showed that curcumin ameliorates nephrosclerosis without affecting hypertension in Dahl rats [34]. Our previous and recent studies also confirmed that curcumin had no significant effect on blood pressure in Dahl rats. Because curcumin has potent antioxidant properties that effectively scavenge reactive oxygen species (ROS), curcumin may exhibit a hypotensive action on ROS-related increase in blood pressure [32,33]. In contrast, the salt-sensitive hypertension in Dahl rats is based on the complex mechanisms [35,36]. An antioxidant, resveratrol, exhibit the therapeutic effect for cardiac dysfunction without affecting blood pressure and cardiac hypertrophy in Dahl rats [37]. Tempol was no effect on hypertension in young Dahl rats [38]. It seems that only antioxidant activity cannot correct salt-sensitive hypertension in Dahl rats. However, curcumin has a protective function against LVH, cardiac dysfunction, and renal dysfunction independent of hypertension, indicating that curcumin in addition to antihypertensive agents such as ARB and ACEI may exhibit synergistic benefits in hypertension-related diseases [39].

  1. Rimbaud, S.; Ruiz, M.; Piquereau, J.; Mateo, P.; Fortin, D.; Veksler, V.; Garnier, A.; Ventura-Clapier, R. Resveratrol improves survival, hemodynamics and energetics in a rat model of hypertension leading to heart failure. PLoS One. 2011, 6(10), e26391.
  2. Vaněčková, I.; Vokurková, M.; Rauchová, H.; Dobešová, Z.; Pecháňová, O.; Kuneš, J.; Vorlíček, J.; Zicha, J. Chronic antioxidant therapy lowers blood pressure in adult but not in young Dahl salt hypertensive rats: the role of sympathetic nervous system. Acta Physiol (Oxf). 2013, 208(4), 340-349.

Comment:

  1. Surprise! surprise!!?? [Page-3: Lines 107- 111 reads "This daily oral administration of curcumin or vehicle was continued for 6 weeks, until the mice reached 12 weeks of age.Systolic blood pressure (SBP) and diastolic blood pressure (DBP) were measured weekly by the tail-cuff method using an indirect blood pressure system (BP-98A; Softron Inc., Japan)."]{The title of the MS reads Rats??}

Response:

              Thank you for your point. We corrected this paragraph in the revised manuscript.

Line 105-114

“Curcumin powder and gum arabic were purchased from FUJIFILM Wako Pure Chemicals Corporation (Osaka, Japan). The DR and DS rats (age, 6-week-old) were randomly assigned to two groups (DR curcumin, n = 5; DS curcumin, n = 8; DR vehicle, n = 5; DS vehicle, n = 8). Rats were daily administered with curcumin (50 mg/kg/d, diluted in 1% arabic gum) and vehicle (1% arabic gum) via an oral gastric gavage. This daily oral administration of curcumin or vehicle was continued for 6 weeks, until the rats reached 12 weeks of age. Systolic and diastolic blood pressure (SBP and DBP) were evaluated by the tail-cuff method using BP-98A system (Softron Inc., Japan).”

Comment:

  1. Page 12: Lines 370-371: [This compound may be used to prevent the development of cardiovascular diseases in patients with hypertension.] [Curcumin's ability to reduce CV diseases is not new in the literature. The real question is - does it delay or prevent? One has to be realistic in using terms.}

Response:

              Thank you for your valuable comment. Your question whether curcumin delay or prevent hypertension-induced LVH is very interesting. Our study could not conclude this question because the treatment period of curcumin continued until the 12 weeks age in Dahl rats. The further experiment is required to clarify this question. We corrected this paragraph in the revised manuscript.

Line 368-373

In conclusion, we have indicated the therapeutic effect of curcumin on hypertension-induced LVH with preserving ejection function through the decrease in acetylated form of GATA4 in Dahl rats. This compound may be used to prevent the development of LVH in patients with hypertension. Future clinical studies are needed to analyze the efficacy of curcumin therapy on prevention of LVH in patients with hypertension.” 

Reviewer 2 Report

The manuscript demonstrated that curcumin functioned as an inhibitor of p300 histone acetyltransferase (HAT) and prevented the development of hypertension-induced left ventricular hypertrophy (LVH). It is evidenced by a decrease in posterior wall thickness and LVH mass, an inhibition in myocardial cell hypertrophy, perivascular fibrosis, and the hypertrophic response gene transcription. The above changes might be due to attenuating the acetylation levels of GATA4. 

The article is well written and easy to read. I have limited comments to the authors that may improve the quality of their excellent work. 

  1. In the discussion, Lines 309-324, please interpret the different findings regarding the effects of curcumin on hypertension. In other words, the author’s study confirmed that curcumin had no significant effect on blood pressure. They also stated the curcumin exhibited a hypotensive function through scavenging reactive oxygen species (ROS). It did not seem very clear to me. Please clarify it.  
  2. Also, please cite the literature,Lines 316-320, not only for transparency but also for permitting readers to look at the article they are interested in. 
  3. On line 339, please spell out the abbreviation HDACs when first shown. 
  4. Legends Figure 5, figure 5A bottom panel should be figure 5D, and figure 5D should be 5E. Please also correct them in the context. 
  5. Lines 209, 271, 309, 325, and 330 have extra spaces at the beginning of the paragraph. 
  6. Remove the letter “b” of acetylationb on line 285. 
  7. The authors concluded that the beneficial effects of curcumin on hypertension-induced LVH via the inhibition of p300-HAT. It would be more convincing if providing those data. 
  8. They observed that the high-salt diet reduced body weight in DS rats. The possible mechanisms might be an interesting question to the readers. Please elaborate on it. 

Author Response

To Review2

Comment:

The manuscript demonstrated that curcumin functioned as an inhibitor of p300 histone acetyltransferase (HAT) and prevented the development of hypertension-induced left ventricular hypertrophy (LVH). It is evidenced by a decrease in posterior wall thickness and LVH mass, an inhibition in myocardial cell hypertrophy, perivascular fibrosis, and the hypertrophic response gene transcription. The above changes might be due to attenuating the acetylation levels of GATA4. 

The article is well written and easy to read. I have limited comments to the authors that may improve the quality of their excellent work. 

Response:

Thank you very much for your helpful critique of our manuscript. In response to your comments, we have provided the revisions of our manuscript:

Comment:

  1. In the discussion, Lines 309-324, please interpret the different findings regarding the effects of curcumin on hypertension. In other words, the author’s study confirmed that curcumin had no significant effect on blood pressure. They also stated the curcumin exhibited a hypotensive function through scavenging reactive oxygen species (ROS). It did not seem very clear to me. Please clarify it.  
  2. Also, please cite the literature,Lines 316-320, not only for transparency but also for permitting readers to look at the article they are interested in. 

Response:

              Thank you for your suggestion. As you mentioned, we added some references and modified this paragraph in the revised manuscript.

Line 304-322

“A natural compound, curcumin, exhibits therapeutic and beneficial potency on cardiovascular diseases, including hypertension. Yao et al. showed that curcumin treatment (300 mg/kg/d) reduced angiotensin II-induced hypertension in mice [32]. N (ω)-nitro-L-arginine methyl ester (L-NAME)-induced hypertension was decreased by curcumin (50 and 100 mg/kg/d) [33]. However, Muta et al. showed that curcumin ameliorates nephrosclerosis without affecting hypertension in Dahl rats [34]. Our previous and recent studies also confirmed that curcumin had no significant effect on blood pressure in Dahl rats. Because curcumin has potent antioxidant properties that effectively scavenge reactive oxygen species (ROS), curcumin may exhibit a hypotensive action on ROS-related increase in blood pressure [32,33]. In contrast, the salt-sensitive hypertension in Dahl rats is based on the complex mechanisms [35,36]. An antioxidant, resveratrol, exhibit therapeutic effect for cardiac dysfunction without affecting blood pressure and cardiac hypertrophy in Dahl rats [37]. Tempol was no effect on hypertension in young Dahl rats [38]. It seems that only antioxidant activity cannot correct salt-sensitive hypertension in Dahl rats. However, curcumin has a protective function against LVH, cardiac dysfunction, and renal dysfunction independent of hypertension, indicating that curcumin in addition to antihypertensive agents such as ARB and ACEI may exhibit synergistic benefits in hypertension-related diseases [39].”

  1. Kong, J.Q.; Taylor, D.A.; Fleming, W.W. Sustained hypertension in Dahl rats. Negative correlation of agonist response to blood pressure. Hypertension. 1995, 25 (1), 139-145.
  2. Karlsen, F.M.; Andersen, C.B.; Leyssac, P.P.; Holstein-Rathlou, N.H. Dynamic autoregulation and renal injury in Dahl rats. Hypertension. 1997, 30 (4), 975-983.
  3. Sunagawa, Y.; Morimoto, T.; Wada, H.; Takaya, T.; Katanasaka, Y.; Kawamura, T.; Yanagi, S.; Marui, A.; Sakata, R.; Shimatsu, A.; Kimura, T.; Kakeya, H.; Fujita, M.; Hasegawa, K. A natural p300-specific histone acetyltransferase inhibitor, curcumin, in addition to angiotensin-converting enzyme inhibitor, exerts beneficial effects on left ventricular systolic function after myocardial infarction in rats. Circ. J. 2011, 75(9), 2151-2159.

Comment:

  1. On line 339, please spell out the abbreviation HDACs when first shown. 

Response:

              Thank you for your point. We corrected this word in the revised manuscript.

Line 336-348

Several studies have implicated that histone de-acetylases (HDAC) and HAT control cardiac hypertrophy [44]. Class IIa HDAC5 and HDAC9 are associated with MEF2, another acetylation target of p300, and regulate its transcriptional activity in response to hypertrophic stimuli. HOP/HDAC2 cooperates with GATA4 and regulates cardiomyocyte proliferation through GATA4 de-acetylation during embryonic development [45]. SIRT7 exerts an anti-hypertrophic action through the deacetylation of GATA4 [46]. In this study, we showed that curcumin reduced the acetylated level of GATA4 in DS rats at the 12-weeks-old. However, the precise mechanism of de-acetylation GATA4 during concentric and pathological hypertrophy is still unclear. Further studies will have to delineate which HDACs regulate the deacetylation of GATA4 during the development of LVH.”

Comment:

  1. Legends Figure 5, figure 5A bottom panel should be figure 5D, and figure 5D should be 5E. Please also correct them in the context. 

Response:

              Thank you for your point. We corrected the legend of Figure 5 in the revised manuscript.

Line 274-20

“To detect the acetylated levels of GATA4, NE the immunoprecipitation with anti-GATA4 antibody were performed. We confirmed the acetylated form of GATA4 was increased in DS rats at the 12 weeks-old (Figure 5D and 5E, lanes 1 and 3, P<0.05). Notably, hemodynamic overload-induced acetylation of GATA4 was reduced by curcumin treatment in DS rats (lanes 3 and 4, P<0.05).”

Line 283-291

Figure 5. Curcumin significantly prevented hemodynamic overload-induced acetylation of GATA4 in DS rats. (A) Nuclear extracts (NE) of LV tissues in each rat were subjected to western blotting with indicated antibodies. (B and C) The protein levels of p300 (B) and GATA4 (C) were normalized by b-actin. (D) The immunoprecipitated samples with anti-GATA4 antibody were subjected to western blotting with indicated antibodies. (E) The acetylated level of GATA4 was normalized to total GATA4. The band intensities were quantified using Image J v1.47 software. The values are represented as the mean ± SEM. All values were normalized to DR vehicle group, which was set at 1.0 as the control.

Comment:

  1. Lines 209, 271, 309, 325, and 330 have extra spaces at the beginning of the paragraph. 
  2. Remove the letter “b” of acetylationb on line 285. 

Response:

              Thank you for your point. We corrected these points in the revised manuscript.

Comment:

  1. The authors concluded that the beneficial effects of curcumin on hypertension-induced LVH via the inhibition of p300-HAT. It would be more convincing if providing those data. 

Response:

              Thank you for your suggestion. We modified the conclusion in the revised manuscript.

Line 368-373

In conclusion, we have indicated the therapeutic effect of curcumin on hypertension-induced LVH with preserving ejection function through the decrease in acetylated form of GATA4 in Dahl rats. This compound may be used to prevent the development of LVH in patients with hypertension. Future clinical studies are needed to analyze the efficacy of curcumin therapy on prevention of LVH in patients with hypertension.”  

Comment:

  1. They observed that the high-salt diet reduced body weight in DS rats. The possible mechanisms might be an interesting question to the readers. Please elaborate on it. 

Response:

              Thank you for your comment. We think that the reduction of body weight in DS rats may be a decrease in the amount of diet due to feeling thirsty, a increases water intake, and a high-salt (8% NaCl). 

Reviewer 3 Report

The manuscript considered the in vivo effect of curcumin in a model of Dahl salt-sensitive rats treated with a high-salt diet, in order to obtain a hypertensive status and evaluate whether curcumin is able to prevent the development of hypertension-induced left ventricular hypertrophy (LVH) with a preserved ejection fraction. The study focused on blood pressure and cardiac function for the considered investigational period; finally the myocardial histology was investigated, as well as the hypertrophy-response gene transcription for ANF and beta-MHC, the expression levels of p300 and GATA4, and the acetylation of GATA4. The results indicate that curcumin does not affect blood pressure or cardiac function, but is able to suppress the hypertension-induced LVH, the expression of cardiac hypertrophy marker genes, and s prevent the acetylation of GATA4. The data complete the previous work of the Authors regarding the cardiac effects of curcumin, and the present results suggest that curcumin may be a useful treatment to be evaluated clinically for the prevention of hypertension-induced LVH.

The study has been carefully conducted, using a well-established rat model of hypertension. Methods are clearly presented, both regarding procedures and variables evaluated; the statistical analysis appears adequately performed, although some aspects may be further addressed (see below). Results are clearly presented and discussed.

Some points to be considered:

Line 48: please define HF at first occurrence of the term.

Methods section: The Authors could include a statement about method of calculation of sample size and justification of sample size (for example, see recommendations in: Charan, Jaykaran, and N D Kantharia. “How to calculate sample size in animal studies?.” Journal of pharmacology & pharmacotherapeutics , 4,4 (2013): 303-6. doi:10.4103/0976-500X.119726 ).

Lines 209-216: reference to the data discussed here (BW, HW, LV, HW/BW, LV/BW) should be given (indicating “Table 2”).  Moreover, here LV refers to “left ventricle weight”; please uniform to the term LVW indicated in Table 2 as well “left ventricle weight”.

Table 1 heading: “at 6 weeks old in Dahl rats”: please improve the syntax. Same for Table 2.

Figure 3E: please make visible the x-axis under bars 1 & 2.

Lines 234-235: “the interstitial fibrosis areas in  each group were comparable (data not shown)”. Do “data not shown” refer to Supplementary Figure S1? In that case, please quote. According to Instructions for Authors (https://www.mdpi.com/journal/nutrients/instructions#suppmaterials), “"Data not shown" should be avoided”.  In addition, the directory named “nutrients-1302780-non-published” should be explained regarding its content, for instance if it is a directory with data available only to the referees just to verify results during the peer-review process.

Lines 368-370: The sentences could be made clearer. In this study it was not directly evaluated the “inhibition of p300-HAT”. Compare also lines 333-337.

Author Response

To Review3

Comment:

The manuscript considered the in vivo effect of curcumin in a model of Dahl salt-sensitive rats treated with a high-salt diet, in order to obtain a hypertensive status and evaluate whether curcumin is able to prevent the development of hypertension-induced left ventricular hypertrophy (LVH) with a preserved ejection fraction. The study focused on blood pressure and cardiac function for the considered investigational period; finally the myocardial histology was investigated, as well as the hypertrophy-response gene transcription for ANF and beta-MHC, the expression levels of p300 and GATA4, and the acetylation of GATA4. The results indicate that curcumin does not affect blood pressure or cardiac function, but is able to suppress the hypertension-induced LVH, the expression of cardiac hypertrophy marker genes, and s prevent the acetylation of GATA4. The data complete the previous work of the Authors regarding the cardiac effects of curcumin, and the present results suggest that curcumin may be a useful treatment to be evaluated clinically for the prevention of hypertension-induced LVH.

The study has been carefully conducted, using a well-established rat model of hypertension. Methods are clearly presented, both regarding procedures and variables evaluated; the statistical analysis appears adequately performed, although some aspects may be further addressed (see below). Results are clearly presented and discussed.

Some points to be considered:

Response:

Thank you very much for your comments on our manuscript. According to your suggestion, we have revised our manuscript:

Comment:

Line 48: please define HF at first occurrence of the term.

Response:

              Thank you for your point. We corrected this word in the revised manuscript.

Line 43-58

Heart failure (HF) is the final stage of cardiovascular diseases and the mortality and morbidity of severe HF are still high [1,2]. Hypertension is one of major risk factors for congestive heart failure and causes left ventricular hypertrophy (LVH) [3]. Chronic pressure and volume overload initially induce LVH with preserved cardiac function. LVH is a compensatory reaction, which eventually lead to maladaptive LVH and pathological remodeling with cardiac dysfunction [3]. This process causes the development of Heart Failure. Hypertensive patients with LVH present a high risk of developing other cardiovascular events, including heart failure [4]. Although effective blood pressure control with the help of agents, such as angiotensin-converting enzyme inhibitors, angiotensin II receptor blockers, and b-blockers, is the primary treatment in patients with hypertension, the overall mortality from this condition is still considerably high [5,6]. Consequently, it is important to identify more effective agents than existing medicines to develop novel pharmacological therapies for HF.”

Comment:

Methods section: The Authors could include a statement about method of calculation of sample size and justification of sample size (for example, see recommendations in: Charan, Jaykaran, and N D Kantharia. “How to calculate sample size in animal studies?.” Journal of pharmacology & pharmacotherapeutics , 4,4 (2013): 303-6. doi:10.4103/0976-500X.119726 ).

Response:

              Thank you for your point. We performed power analysis using G*Power software and calculated sample size in this study. We described the calculated data in the revised manuscript as follows:

Line 164-171

Power analysis was performed using G*Power 3.197 software Based on our previous study [20], we set an effect size of 0.4, a power of 0.8, statistical significance of 0.05, and expected attrition of 10%, and determined that a sample size was ten rats per DS group. The data are expressed as the mean ± standard error (SE). Data analysis were performed using the StatView 5.0 software by two-way analysis of variance, following by Tukey’s multiple-comparison test. A value of P < 0.05 is statistical ignificance.”

Comment:

Lines 209-216: reference to the data discussed here (BW, HW, LV, HW/BW, LV/BW) should be given (indicating “Table 2”).  Moreover, here LV refers to “left ventricle weight”; please uniform to the term LVW indicated in Table 2 as well “left ventricle weight”.

Response:

              Thank you for your point. We corrected “left ventricle weight (LVW)“ at the section of “Results”, heading at Table 2 in the revised manuscript.

Line 212-219

While the high-salt diet reduced body weight (BW) in DS rats, these values were also similar in DS vehicle and DS curcumin groups. Hypertension induced an increase in heart weight (HW), LV weight (LVW), the HW/BW ratio and the LVW/BW ratio. Curcumin treatment significantly reduced these parameters without affecting blood pressures in DS rats. These results indicated that curcumin treatment significantly prevented hypertension-induced LVH without affecting blood pressure and cardiac function in DS rats.”

Comment:

Table 1 heading: “at 6 weeks old in Dahl rats”: please improve the syntax. Same for Table 2.

Response:

              Thank you for your point. We corrected headings of Table 1 and 2 in the revised manuscript.

Table 1, heading,

Table 1. Hemodynamic parameters in Dahl rats at 6-weeks-old

Table 2, heading,

Table 2. BW, HW, LVW, and blood pressure parameters in Dahl rats at 12-weeks-old

Comment:

Figure 3E: please make visible the x-axis under bars 1 & 2.

Response:

              Thank you for your point. We corrected this graph in the revised manuscript.

Comment:

Lines 368-370: The sentences could be made clearer. In this study it was not directly evaluated the “inhibition of p300-HAT”. Compare also lines 333-337.

Response:

              Thank you for your comment. As you mentioned, we modified this paragraph in the revised manuscript as follows:

Line 368-373

In conclusion, we have indicated the therapeutic effect of curcumin on hypertension-induced LVH with preserving ejection function through the decrease in acetylated form of GATA4 in Dahl rats. This compound may be used to prevent the development of LVH in patients with hypertension. Future clinical studies are needed to analyze the efficacy of curcumin therapy on prevention of LVH in patients with hypertension.”

Comment:

Lines 234-235: “the interstitial fibrosis areas in each group were comparable (data not shown)”. Do “data not shown” refer to Supplementary Figure S1? In that case, please quote. According to Instructions for Authors (https://www.mdpi.com/journal/nutrients/instructions#suppmaterials), “"Data not shown" should be avoided”.  In addition, the directory named “nutrients-1302780-non-published” should be explained regarding its content, for instance if it is a directory with data available only to the referees just to verify results during the peer-review process.

Response:

              Thank you for your important comment. We mentioned the interstitial fibrosis areas in each group in the Supplementary Figure S1. “nutrients-1302780-non-published” includes the raw images of H&E-staining of whole hearts and myocardial cell and the results of echocardiography. As you suggested, we modified this paragraph in the revised manuscript as follows:

Line 222-240

“To investigate whether curcumin affected hypertension-induced myocardial cell hypertrophy and fibrosis, we performed histological analysis in all rats. The representative images of H&E staining of transversely sectioned LV and myocardial cells in each group are shown in Figure 3A and 3B. The data of myocardial cell diameter in each group are shown in Figure 3C. Microscopic analysis revealed that hypertension induced an increase in myocardial cell diameter in DS vehicle group compared to that in DR vehicle group (lanes 1 and 3, P<0.001). Curcumin treatment significantly reduced this increase compared to vehicle treatment in DS rats (lanes 3 and 4, P<0.01). The representative images of perivascular fibrosis in each group are shown in Figure 3D. The ratio of perivascular fibrosis area around the coronary artery was significantly increased in DS vehicle group compared to in DR vehicle group (Figure 3E, lanes 1 and 3, P<0.001). Hypertension-induced perivascular fibrosis was significantly inhibited by curcumin treatment (lanes 3 and 4, P<0.01). The areas of interstitial fibrosis in each group were comparable (Supplemental Figure S1). These indicated that curcumin treatment significantly suppressed hypertension-induced increases in myocardial cell diameter and perivascular fibrosis in DS rats.”

Round 2

Reviewer 1 Report

Minor revisions needed- Please get this MS re-read by someone familiar with scientific english... I found several minor trouble spots....please scan carefully again ! Thank you!

Lines 80- 84:  (It should be) Even though curcumin has low bioavailability, highly bioavailable forms of curcumin is expected to exhibit potential health benefits and expected to contribute to the health management [21,22].

Line 83: (It should be) Our previous study has shown...

Lines 216-217: (it should be)  Curcumin treatment significantly reduced these parameters without affecting blood pressure in DS rats

Line 225: (It should be) The representative images of H&E stained of

Lines 305 - 308: (It should be) Curcumin, a natural compound, exhibits therapeutic and potentially beneficial effects on cardiovascular diseases, including hypertension.

The authors have mentioned 'prevention' of fibrosis several times. Is it prevention or reduction or decrease the intensity....please make sure of this!

Author Response

Reviewer 1:

Minor revisions needed- Please get this MS re-read by someone familiar with scientific english... I found several minor trouble spots....please scan carefully again ! Thank you!

Response:

Thank you for your valuable suggestion. In accordance with your comments, we corrected them in the revised manuscript.

Comment:

Lines 80- 84:  (It should be) Even though curcumin has low bioavailability, highly bioavailable forms of curcumin is expected to exhibit potential health benefits and expected to contribute to the health management [21,22].

Line 83: (It should be) Our previous study has shown...

Lines 216-217: (it should be)  Curcumin treatment significantly reduced these parameters without affecting blood pressure in DS rats

Line 225: (It should be) The representative images of H&E stained of

Lines 305 - 308: (It should be) Curcumin, a natural compound, exhibits therapeutic and potentially beneficial effects on cardiovascular diseases, including hypertension.

Response:

Thank you for your points. We corrected them in the revised manuscript as follow:

Line 80-84

“Even though curcumin has low bioavailability, highly bioavailable forms of curcumin is expected to exhibit potential health benefits and expected to contribute to the health management [21,22]. Our previous study has shown that curcumin prevents systolic dysfunction and the development of myocardial cell hypertrophy via the inhibition of p300-HAT activity in two HF models in rats, with MI and hypertension [23].”

Line 215-219

“Curcumin treatment significantly reduced these parameters without affecting blood pressure in DS rats. These results indicated that curcumin treatment significantly prevented hypertension-induced LVH without affecting blood pressure and cardiac function in DS rats.”

Line 225

“The representative images of H&E stained of transversely sectioned LV and myocardial cells in each group are shown in Figure 3A and 3B.”

Line 303-304

“Curcumin, a natural compound, exhibits therapeutic and beneficial potency on cardiovascular diseases, including hypertension.”

Comment:

The authors have mentioned 'prevention' of fibrosis several times. Is it prevention or reduction or decrease the intensity....please make sure of this!

Response:

Thank you for your indication. We described as curcumin “reduced” fibrosis in the revised manuscript.

Line 23-40

Abstract: We found that curcumin, a p300 histone acetyltransferase (HAT) inhibitor, prevents cardiac hypertrophy and systolic dysfunction at the stage of chronic heart failure in Dahl salt-sensitive rats (DS). It is unclear whether curcumin suppresses the development of hypertension-induced left ventricular hypertrophy (LVH) with a preserved ejection fraction. Therefore, in this study, we randomized DS (n = 16) and Dahl salt-resistant (DR) rats (n = 10) at 6 weeks of age to either curcumin or vehicle groups. These rats were fed a high-salt diet and orally administrated with 50 mg/kg/d curcumin or its vehicle for 6 weeks. Both curcumin and vehicle treatment groups exhibited similar degrees of high-salt diet-induced hypertension in DS rats. Curcumin significantly decreased hypertension-induced increase in posterior wall thickness and LV mass index, without affecting the systolic function. It also significantly reduced hypertension-induced increases in myocardial cell diameter, perivascular fibrosis, and transcriptions of the hypertrophy-response gene. Moreover, it significantly attenuated the acetylation levels of GATA4 in the hearts of DS rats. A p300 HAT inhibitor, curcumin, suppresses the development of hypertension-induced LVH, without affecting blood pressure and systolic function. Therefore, curcumin may be used for the prevention of development of LVH in patients with hypertension.”

Line 235-243

“Hypertension-induced perivascular fibrosis was significantly reduced by curcumin treatment (lanes 3 and 4, P<0.01). The areas of interstitial fibrosis in each group were comparable (Supplemental Figure S1). These indicated that curcumin treatment significantly reduced hypertension-induced increases in myocardial cell diameter and perivascular fibrosis in DS rats.

Figure 3. The hemodynamic overload-induced concentric left ventricular hypertrophy (LVH), myocardial cell hypertrophy, and perivascular fibrosis were significantly reduced by curcumin treatment in DS rats.”

Line 362-367

“It is possible that the inhibitory effects of curcumin against oxidative stress and inflammatory process may reduce hemodynamic overload-induced perivascular fibrosis. Therefore, it is important to clarify the specific effects of curcumin on cardiac fibroblasts as well as its role on HF model animals in future studies.”
